# A Review of Alternative Procedures to the Bond Ball Mill Standard Grindability Test

**Vladimir Nikolić** [1], **Gloria G. García** [2], **Alfredo L. Coello-Velázquez** [3], **Juan M. Menéndez-Aguado** [2,*], **Milan Trumić** [1] **and Maja S. Trumić** [1]

1    Technical Faculty in Bor, University of Belgrade, 19210 Bor, Serbia; vnikolic@tfbor.bg.ac.rs (V.N.); mtrumic@tfbor.bg.ac.rs (M.T.); majatrumic@tfbor.bg.ac.rs (M.S.T.)
2    Escuela Politécnica de Mieres, University of Oviedo, Gonzalo Gutiérrez Quirós, 33600 Mieres, Spain; gloria.glez.gcia@gmail.com
3    CETAM, Universidad de Moa Antonio Núñez Jiménez, Moa 83300, Cuba; acoello@ismm.edu.cu
\*    Correspondence: maguado@uniovi.es; Tel.: +34-985458033

**Abstract:** Over the years, alternative procedures to the Bond grindability test have been proposed aiming to avoid the need for the standard mill or to reduce and simplify the grinding procedure. Some of them use the standard mill, while others are based on a non-standard mill or computation techniques. Therefore, papers targeting to propose a better alternative claim to improve validity, to reduce test duration, or to propose simpler and faster alternative methods for determining the Bond work index ($w_i$). In this review paper, a compilation and critical analysis of selected proposals is performed, concluding that some of the short procedures could be useful for control purposes, while the simulation-based procedures could be interesting within a process digitalisation strategy.

**Keywords:** grindability; comminution; Bond work index

## 1. Introduction

Determining the Bond index using the Fred Bond method [1,2] is considered the state-of-the-art methodology for mill calculations and a critical process parameter in raw materials selection and grinding process control. Although it is usually referred to as a standard test, no ISO (International Organization for Standardization) or ASTM (American Society for Testing and Materials) standard procedure has been established, so the primary reference used worldwide to define the procedure is the original proposal from Bond. Despite this, the knowledge of the Bond standard test is enriched continuously with new research, as is the case of the recently published work by García et al. [3], which presents a deep analysis of the test procedure and evidences the importance of the grindability index (proposing it to be renamed as the Maxson index), or the recent proposal by Nikolić and Trumić [4], which represent a new approach for determination Bond work index on finer samples.

Alternative tests soon arose after Bond's proposal to avoid the need for the standard mill and time-consuming procedure. Therefore, papers dealing with this problem are numerous, aiming to discuss the validity of simpler and quicker methods to determine the Bond work index ($w_i$). Some of them use the standard mill, while others use a non-standard mill or are based on computation techniques. In this review paper, a compilation and critical analysis of several selected methodologies were performed, based on the practical experience of the laboratories involved in this research.

It is worthy to mention the development of other approaches to grindability evaluation based on impact breakage tests. The drop weight test has proven its validity and scaling-up possibilities under certain conditions [5–7].

There are not many review papers describing alternative methods of ball mill $w_i$ determination. The work of Lvov and Chitalov [8] is probably the most recent one, and

it performs a sound analysis of several alternative methodologies. This review includes additional methodologies and considers the analysis of the relative square error and the procedure advantages claimed by the authors of each proposal.

## 2. Alternative Procedures to the Bond Ball Mill Standard Test

Berry and Bruce [9] introduced the first alternative procedure to the Bond standard test. The procedure is based on determining the grindability of an unknown ore by comparing it to the grindability behaviour of a reference ore. It can be performed in any laboratory ball mill, but it requires a reference sample ore for which $w_i$ is known. In the Berry and Bruce procedure, 2 kg weight samples of the reference and unknown ores with a particle size under 1.651 mm are wet ground in a laboratory ball mill that is 305 mm in diameter, using active power monitoring. According to the Bond Third's Law of comminution (Equation (1); [1]), after performing both grinding tests with the same specific active power energy consumption, Equation (2) can be deduced:

$$W = 10 \cdot w_i \cdot \left[ \frac{1}{\sqrt{P_{80}}} - \frac{1}{\sqrt{F_{80}}} \right] \ [kWh/t] \tag{1}$$

$$w_{ir} \cdot \left[ \frac{1}{\sqrt{P_r}} - \frac{1}{\sqrt{F_r}} \right] = w_{i,BB} \cdot \left[ \frac{1}{\sqrt{P_{80}}} - \frac{1}{\sqrt{F_{80}}} \right] \Rightarrow w_{i,BB} = w_{ir} \cdot \frac{\left[ \frac{1}{\sqrt{P_r}} - \frac{1}{\sqrt{F_r}} \right]}{\left[ \frac{1}{\sqrt{P_{80}}} - \frac{1}{\sqrt{F_{80}}} \right]}, \ [kWh/t] \tag{2}$$

wherein:

$W$—Specific power consumption, (kWh/t);
$w_{ir}$—Bond work index of reference ore, (kWh/t);
$P_r$—80% passing product particle size, reference ore, (μm);
$F_r$—80% passing feed particle size, reference ore, (μm);
$w_{i,BB}$—Bond work index estimation of the unknown ore, (kWh/t);
$P_{80}$—the 80% passing product particle size, unknown ore, (μm);
$F_{80}$—the 80% passing feed particle size, unknown ore, (μm).

The validity of this procedure depends on the accuracy of stopping the unknown sample grinding test after a specific power consumption is reached (measured with a power-meter) and on the similarity of the particle size distribution (PSD) of the feed samples. Differences in sample densities and PSD affect the density and rheological characteristics of the pulp when performing the test wet way. Moreover, it has been proven that $w_i$ is not a constant value for each ore, so the reference sample value would only be valid within a specific grinding size range [3,10]. The main advantage of this procedure is that it is fast and does not require Bond's standard ball mill, but accurate power measurements are needed, and the use of a reference ore as if $w_i$ had a constant value is also a source of inaccuracy.

Horst and Bassarear [11] gave a procedure based on Berry and Bruce [9], but with a basis on grinding kinetics. In this case, the procedure does not consider the unknown sample feed and grinding product PSD; instead, starting from the reference ore feed PSD, the unknown ore grinding product PSD is calculated by a first-order kinetics equation (Equation (3)):

$$R = R_0 \cdot e^{-k \cdot t} \tag{3}$$

wherein:

$R$—oversize of the comparative sieve after grinding time $t$;
$R_0$—oversize of the comparative sieve at the beginning of grinding $t = t_0 = 0$;
$k$—first-order kinetics grinding constant;
$t$—grinding time.

The test can be performed in any laboratory ball mill on a sample with an initial size under 1.651 mm. A reference ore sample weighing 1 kg is ground until the desired mill PSD is obtained. This can be performed with several grinding tests in a row on the same sample, accumulating the grinding time from one test to another; the sample PSD is obtained after

each grinding test, and if a finer product is needed, the sample is returned to the mill and ground for more time. Three unknown ore samples weighing 1 kg are ground in the same mill under the same conditions and with different grinding times. The grinding times of these three samples should include the grinding time of the reference ore. The PSD is determined for all grinding products, and a plot $t$ versus $lnR$ can be performed to obtain the grinding rate constants $k_i$ for each grinding size. The unknown ore grinding product can be calculated using the reference ore feed PSD and the grinding time of the reference ore to the desired fineness, and the value $k_i$ is determined (Equation (3)) in the cycle of ore grindability. Based on the PSD calculated in this way, the value of the parameter $P_{80}$ (μm) is determined, and the value of the parameter $F_{80}$ (μm) is taken to be equal to the parameter $F_r$. The Bond work index is estimated by Equation (4).

$$w_{i,HB} = w_{ir} \cdot \frac{\left[ \frac{1}{\sqrt{P_r}} - \frac{1}{\sqrt{F_r}} \right]}{\left[ \frac{1}{\sqrt{P_{80}}} - \frac{1}{\sqrt{F_r}} \right]}, \ [kWh/t] \tag{4}$$

Differences in grindability in this process are reflected only through differences in the size of the grinding product $P_{80}$. The advantages of this procedure are the use of an ordinary laboratory mill with balls and a smaller mass and sample size than the standard Bond test. A small amount of time is needed to perform the test and calculate the PSD. The total execution time of this test is almost no shorter than the standard Bond test. The Berry–Bruce procedure, which is similar to this test, is considerably shorter and should give more reliable results. However, the mean square relative difference reported by the authors between the values of the Bond work index obtained by the standard method and the values obtained by the Berry–Bruce method is 8.25% and 1.72%. for the Horst–Bassarear method. The relative difference achieved by the Horst–Bassarear procedure is surprisingly small, although the PSD of the feed sample is equated to the PSD of the reference ore feed sample, and the PSD of the unknown ore grinding product is calculated using the grinding kinetics equation.

Smith and Lee [12] determined the Bond work index in a standard mill for eight different materials at different openings of a comparative sieve according to the standard Bond test. They compared the data obtained by the standard Bond test and the data from the open-circuit grinding, i.e., the first grinding cycle of the standard Bond test. The tests showed that the parameter $G_z$ [g/rev] of the last grinding cycle of the standard Bond test and the parameter $G_0$ of the open-circuit grinding under the same conditions are in a direct correlation $G_0 = f(G_z)$. This correlation was established on screens with smaller openings and in tests performed with less than 300 mill revolutions. With this correlation, it is possible to estimate $G_z$ in the standard Bond test based on the value of $G_0$ determined in the open circuit grinding, and the estimated Bond work index ($w_{i,SL}$) can then be calculated according to Equation (5).

$$w_{i,SL} = 1.1 \cdot \frac{16}{G^{0.82}} \cdot \sqrt{\frac{P_{100}}{100}} \ [kWh/t] \tag{5}$$

A correlation that is established in this way is valid only for the materials on which it is determined. For other materials, it is necessary to establish a new correlation relationship, which requires a Bond mill and sample preparation conducted in the same way as the standard Bond test. A lot of work and grinding cycles are needed to determine the correlation $G_z = K \cdot G_0$. The Bond work index is estimated based on one grinding cycle performed in the standard mill and calculated by following Equation (5). The Smith–Lee results showed that the differences from the standard Bond test and the $w_{i,SL}$ values do not exceed 15%. Probably one of the main shortcomings of this methodology is the influence of feed particle size on the initial cycles, which could be the main source of deviation.

Kapur [13] analysed the grinding cycles that made up the standard Bond test using a mathematical algorithm based on first order grinding kinetics and concluded that the

estimation of $w_i$ could be performed based on the results of the first two grinding cycles from the standard Bond test. In several tests using different materials, Kapur observed no significant difference between the grinding rate constant of classes above $P_k$ from a fresh sample and the circulating load in the standard Bond test. He suggested that the grinding rate constant from the second grinding cycle of the standard Bond test could be used to estimate the $w_i$ using Equation (6):

$$w_{i,K} = 1.1 \cdot 2.648 \cdot P_{100}{}^{0.406} \cdot k_2{}^{-0.81} \cdot (X \cdot M)^{-0.853} \cdot (1-X)^{-0.099}, (kWh/t) \tag{6}$$

wherein:

$P_{100}$—closing sieve size, (μm);

$k_2$—grinding rate constant of class $+P_k$ from the second grinding cycle of the standard Bond test:

$$k_2 = \frac{\ln[M - Z_1 \cdot (1-X)] - \ln(M - Z_2)}{N_2} \tag{7}$$

wherein:

$X$—participation of size class over $P_{100}$ in the initial sample, (partial unit);

$M$—mineral charge in the mill, (g);

$Z_1$ and $Z_2$—weight of the under-size in the first and grinding cycle, (g);

$N_2$—number of revolutions of the mill in the second grinding cycle, (rev).

Numerical coefficients and exponents in Equation (6) were determined using the least-squares method, provided that the differences between the estimated and experimental values of $w_i$ were minimised. The mean square relative error reported between the values of the standard method $w_i$ and $w_{i,K}$ was 9.7%. Kapur stated that this abbreviated test does not substitute for the standard Bond test, recommending it for daily ore grinding monitoring for control purposes

Karra [14] developed a mathematical algorithm for simulating the Bond test based on the first two grinding cycles from the standard test. It can be considered a modified procedure of the one proposed by Kapur [13]. He considered that the circulating load in the standard Bond test has lower grindability and shows slower grinding behaviour. The Bond test is simulated until a circulating load of 250% is established. The value $G$ (g/rev) is obtained from the last simulated grinding cycle, but $P_{80}$ (μm) cannot be estimated. Therefore, in this procedure, the Bond formula cannot be used to calculate the work index, but the empirical formula obtained by statistical data processing can be used. The Karra algorithm is performed using the first two cycles of the standard Bond test and then determining the estimated value:

$M$—sample mass, (g);

$C = \frac{M}{3.5}$—desired under-size mass of the closing screen size at steady state, (g);

$F_{80}$—80% passing feed particle size, (μm);

$Y$—class participation $(-P_{100} + 0)$ in the starting sample, (partial unit);

$Z_1$ and $Z_2$—weight of the under-size of the closing screen size in the first and second grinding cycle, (g);

$N_1$ and $N_2$—number of mill revolutions in the first and second grinding cycle.

Further simulation is performed by calculation, provided that $M \cdot Y < C$, according to the following formulas:

$$k_1 = \frac{(1-Y)}{N_1} \cdot \left( \frac{Z_1 - M \cdot Y}{M - M \cdot Y} \right) \tag{8}$$

$$k_2 = \frac{1}{(M - Z_1) \cdot N_2} \cdot (Z_2 - Z_1 \cdot Y - Z_1 \cdot k_1 \cdot N_2) \tag{9}$$

First cycle:

$$G_1 = \frac{Z_1 - M \cdot Y}{N_1} \tag{10}$$

Second cycle:

$$G_2 = \frac{Z_2 - Z_1 \cdot Y}{N_2} \tag{11}$$

Subsequent cycles:

$$N_i = \frac{C - Y \cdot Z_{i-1}}{G_{i-1}} \tag{12}$$

$$Z_i = Z_{i-1} \cdot Y + Z_{i-1} \cdot N_i \cdot k_1 + (M - Z_{i-1}) \cdot N_i \cdot k_2 \tag{13}$$

$$G_i = \frac{Z_i - Y \cdot Z_{i-1}}{N_i} \tag{14}$$

The simulation is performed until a stable value of $G$ (g/rev) is reached. The Bond work index is estimated by Equation (15).

$$w_{i,Kr} = 1.1 \cdot 9.934 \cdot P_c^{0.308} \cdot G^{-0.696} \cdot F_{80}^{-0.125}, \ [\text{kWh/t}] \tag{15}$$

wherein:

$P_c$—closing screen size, (μm);

$G$—net weight of undersize product per unit revolution of the mill, (g/rev);

$F_{80}$—the 80% passing feed particle size, (μm).

The mean square relative error between $w_i$ and $w_{i,Kr}$ is 4.77%, better than the Kapur algorithm.

Mular and Jergensen [15] proposed the Anaconda method, which does not require a Bond mill or a reference ore for comparison in each test. The Anaconda procedure uses a mill that is calibrated with a reference ore or ores, and the Bond work index is calculated by Equation (16):

$$w_{i,An} = \frac{A}{\left( \frac{1}{\sqrt{P_{80}}} - \frac{1}{\sqrt{F_{80}}} \right)}, \ [\text{kWh/t}] \tag{16}$$

where in:

$A$—mill calibration factor, (kWh/t);

$F_{80}$—the 80% passing feed particle size, (μm);

$P_{80}$—product on milling which grindability is determined, (μm).

To determine the calibration constant $A$ of the laboratory mill, the value of the work index $w_i$ at a given size of the opening of the closing sieve size $P_{100}$ should be determined on the reference ore(s) by the standard Bond test. After that, samples of the same ores should be ground in the laboratory mill at the same time $t$ and determined for each grinding cycle $F_{80}$ and $P_{80}$. Based on the obtained results, the mill calibration constant $A$ is determined as the average value of several measurements using Equation (17).

$$A = w_i \cdot \left( \frac{1}{\sqrt{P_{80}}} - \frac{1}{\sqrt{F_{80}}} \right) \tag{17}$$

At the Anaconda Research Center, they worked with a mill 210 mm in diameter and 251 mm long at 96% of critical speed and charged with the grinding media distribution shown in Table 1.

The feed consisted of 1 kg samples with the particle size (−1.651 + 0.147 mm). The closing sieve size was $P_{100}$ = 147 μm; wet grinding was performed with 50% wt solids in the pulp for 10 min. Under these conditions, they reported $A$ = 0.5031 kWh/t, so Equation (16) could be written as shown in Equation (18).

$$w_{i,An} = \frac{0.5031}{\left( \frac{1}{\sqrt{P_{80}}} - \frac{1}{\sqrt{F_{80}}} \right)}, \ [kWh/t] \tag{18}$$

$A$ value varies with $P_{100}$, the feed weight, the grinding time, and other grinding parameters. Equation (18) gives a work index estimation for $P_{100}$ = 147 μm under the

grinding conditions at the Anaconda Research Center. The mean square relative error between $w_i$ and $w_{i,An}$ was reported as 4.09%., which can be considered as excellent. The procedure itself is quick and straightforward when $A$ is known, although its determination must be performed carefully.

**Table 1.** Ball loading of the mill used in the Anaconda method.

| Diameter of Balls, mm | Number of Balls | Mass, g |
|:---:|:---:|:---:|
| 35.6–38.1 | 11 | 2316.5 |
| 31.8–33.0 | 17 | 2325.4 |
| 29.2–31.0 | 13 | 1534.8 |
| 25.4–27.9 | 10 | 822.5 |
| 24.1–25.4 | 7 | 449.7 |
| 22.9–24.1 | 30 | 1634.0 |
| Total | 88 | 9082.9 |

Nematollahi [16] proposed the estimation of $w_i$ using a 200 mm $\times$ 200 mm mill with the grinding charge shown in Table 2.

**Table 2.** Characteristics of balls used by Nematollahi in the test.

| Ball diameter (mm) | 38.1 | 31.75 | 25.4 | 19.05 | 15.87 |
|:---:|:---:|:---:|:---:|:---:|:---:|
| Number of balls | 13 | 20 | 3 | 21 | 28 |

The initial sample volume is 207 cm$^3$ instead of 700 cm$^3$; accordingly, the test can be performed on 3 kg instead of 10 kg. The procedure involves dry grinding in a closed cycle until a 250% circulating load is reached. The Bond work index is estimated using Equation (19).

$$w_{i,N} = \frac{11.76}{p_{100}^{0.23}} \cdot \frac{1}{G^{0.75}} \cdot \frac{1}{\frac{10}{\sqrt{P_{80}}} - \frac{10}{\sqrt{F_{80}}}} \tag{19}$$

The main advantage is the lower sample mass requirement. The disadvantage of this procedure is the calibration of the mill itself. Table 3 shows the comparative values obtained between the standard Bond ball mill and the Nematollahi mill.

Menéndez-Aguado et al. [17] examined the possibility of determining the work index in a Denver laboratory batch ball mill (Figure 1) with the same inner diameter as the Bond ball standard mill. The research was performed on the size class of 100% −3.35 mm using samples of gypsum, celestite, feldspar, clinker, limestone, fluorite, and copper slag. Considering that the Bond mill/Denver mill volume ratio is 2.15, the initial sample volume was 326 cm$^3$ instead of 700 cm$^3$. Accordingly, the grinding charge was adjusted, as shown in Table 4. The grinding procedure in the Denver mill followed Bond's methodology step by step, only needing volume adjustment. The Bond work index is estimated by following Equation (20):

$$w_{i,MA} = \frac{44.5}{p_{100}^{0.23} \cdot (2.15 \cdot G)^{0.82} \cdot \left( \frac{10}{\sqrt{P_{80}}} - \frac{10}{\sqrt{F_{80}}} \right)} \tag{20}$$

**Table 3.** Comparative values of $w_i$ and $w_{i,N}$.

| Sample | Bond Mill $w_i$ (kWh/t) | Nematollahi Mill $w_{i,N}$ (kWh/t) | Difference (%) |
|---|---|---|---|
| Barite | 6.12 | 6.21 | 1.47 |
| Feldspar | 11.75 | 11.12 | −5.36 |
| Hematite | 13.89 | 14.31 | 3.02 |
| Calcite | 8.36 | 8.50 | 1.67 |
| Chromite | 14.98 | 15.70 | 4.81 |
| Dolomite | 21.77 | 19.18 | −11.90 |
| Coke | 30.43 | 28.75 | −5.52 |
| Coal | 12.99 | 12.33 | −5.08 |
| Silica | 11.93 | 11.49 | −3.69 |
| Fluorite | 7.40 | 7.28 | −1.62 |
| Magnetite | 9.33 | 9.54 | 2.25 |
| Mean-square relative error | | | 5.09 |

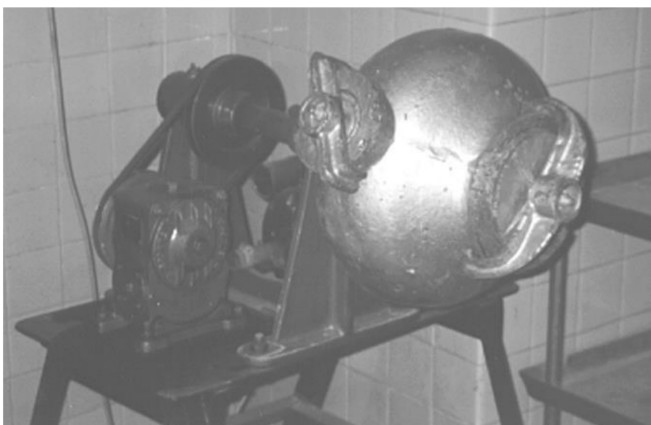

**Figure 1.** Denver laboratory batch ball mill.

**Table 4.** Ball charge in Bond and Denver mills used by Menendez Aguado et al.

| Bond Mill (1952) | | | Denver Mill | | |
|---|---|---|---|---|---|
| Number of Balls | Diameter, cm | Mass, g | Number of Balls | Diameter, cm | Mass, g |
| 22 | 3.810 | 5951 | 10 | 3.810 | 2705 |
| 34 | 3.175 | 4767 | 16 | 3.175 | 2243 |
| 50 | 2.540 | 3750 | 23 | 2.540 | 1725 |
| 54 | 2.223 | 3007 | 25 | 2.223 | 1393 |
| 73 | 1.905 | 2920 | 34 | 1.905 | 1360 |
| Total: 233 | Total: | 20396 | Total:108 | Total: | 9426 |

The main advantages of this procedure are the availability of the Denver mill and the lower initial mass requirement. Table 5 shows the comparative values reported, showing a mean square relative error of 3.71%.

**Table 5.** Comparative values of $w_i$ and $w_{i,MA}$.

| Sample | $P_{100}$ (μm) | $w_i$ (kWh/t) | $w_{i,MA}$ (kWh/t) | Difference (%) |
|---|---|---|---|---|
| Limestone | 200 | 8.99 | 9.05 | −0.67 |
| Feldspar | 200 | 11.06 | 10.92 | +1.27 |
| Celestite | 200 | 5.41 | 5.57 | −2.96 |
| Clinker | 200 | 12.36 | 12.25 | +0.89 |
| Gypsum | 200 | 6.08 | 5.78 | +4.93 |
| Fluorspar | 200 | 6.94 | 7.41 | −6.77 |
| Copper slag | 200 | 18.40 | 19.10 | −3.80 |
| Mean-square relative error | | | | +3.71 |

Mucsi [18] presented a relatively fast method for estimating $w_i$ for brittle materials (limestone, crushed pebble, bauxite, zeolite and basalt) using a Hardgrove mill with a torque meter (measuring cell), enabling the direct measurement of the power delivered to the mill. The test requires 50 g of the initial sample size of 1180–600 μm, the load of the ring on the grinding tip to be 290 N, and the grinding time to be 3 min (60 revolutions of the mill at a speed of 20 rpm). The closing screen size is 75 μm, and the Hardgrove index is determined by Equation (21):

$$H = 13 + 6.93 \cdot m_H \tag{21}$$

wherein:

$m_H$—weight under 75 μm;
$H$—Hardgrove index.
The Bond work index can be estimated from the Hardgrove index using Equation (22):

$$w_{i,H} = \frac{435}{H^{0.82}} \tag{22}$$

The Hardgrove index is based on fine products for a given number of revolutions, and the Bond work index is based on the mass of a fine product multiplied by the number of revolutions of the mill. However, it should be taken into account that other factors may also influence the given input torque in the grinding mill, such as friction, cohesion, adhesion, and material volume flow characteristics. These parameters are taken into account when specific grinding is measured by measuring torque in the manner described previously. Specific energy consumption ($W_{s,H}$) is calculated using Equation (23) when grinding is performed in a universal Hardgrove mill. The measurement of no-load energy (torque) must be subtracted from the total measured energy to determine only the specific shredding energy.

$$W_{s,H} = \frac{\int_0^\tau 2\pi n[M(t) - M_0]dt}{m} \tag{23}$$

wherein:

$M(t)$—torque (balls + material) (Nm);
$M_0$—no-load torque (Nm);
$n$—revolution per min (1/s);
$t$—grinding time (s);
$m$—mass of sample below 75 μm (g).
The specific energy consumption for comminution in the Bond mill is calculated by Equation (24);

$$W_{s,B} = \frac{W_{balls+material} - W_0}{m_p} \tag{24}$$

wherein:

$W_{balls+material}$—measured work with ball and mineral charge (kWh);
$W_0$—measured work without charge (kWh);
$m_p$—the product mass (t).

Once $W_{s,B}$ or $W_{s,H}$, $P_{80}$, and $F_{80}$ are known, $w_i$ can be estimated after isolating it in Equation (1) for both cases.

The reported relative difference values between $w_i$ and $w_{i,H}$ for different ores ranged from −8.1 to 24.1%. The advantages of this method are the use of a simple well-controlled laboratory mill, the need of only 50 g of sample, and a short testing time (60–90 min).

Saeidi et al. [19] rely on the mill designed by Nematollahi [16] to determine the Bond work index, estimated by Equation (19). Using a representative sample (iron ore) of 2 kg, the PSD was determined, and the representative sample was then ground at time intervals of 20, 60, 120, and 180 s. After each grinding, the sample was sieved, $P_{80}$ was obtained, and the work index was determined. The obtained results are shown in Table 6.

**Table 6.** Comparative results for $w_i$ versus $w_{i,N}$.

| Grinding Time (s) | $w_i$ (kWh/t) | $w_{i,N}$ (kWh/t) | Difference (%) |
|---|---|---|---|
| 20 | 13.1 | 8.21 | 37.33 |
| 60 | 12.36 | 8.04 | 34.95 |
| 120 | 11.68 | 7.84 | 32.88 |
| 180 | 11.11 | 7.81 | 29.70 |

Due to large deviations shown in Table 6, the authors defined a new Equation (27) to determine the Bond work index based on the obtained results. They came up with a new formula by examining the relationship between the parameters $G$ (g/rev) and $P_{80}$ (μm) and the grinding time for this ore, resulting in Equations (25) and (26):

$$P_{80} = -0.1085 \cdot t + 122.56 \tag{25}$$

$$G = -1E - 0.6 \cdot t^2 + 0.0004 \cdot t + 0.3397 \tag{26}$$

Finally, $w_i$ can be estimated using Equation (27), which is the result of combining Equation (25) and Equation (26) with Equation (19), resulting in a new equation to estimate the Bond work index:

$$w_{i,SA} = \frac{5.6}{(-1E - 0.6 \cdot t^2 + 0.0004 \cdot t + 0.3397)^{0.75}} \cdot \frac{1}{\frac{10}{\sqrt{-0.1085 \cdot t + 122.56}} - \frac{10}{\sqrt{F_{80}}}} \tag{27}$$

In order to determine the accuracy of Equation (27), an additional grinding run of 100 s was performed, and the results of which are shown in Table 7.

**Table 7.** Comparison of results for $w_i$ versus $w_{i,SA}$.

| Grinding Time (s) | $w_i$ (kWh/t) | $w_{i,SA}$ (kWh/t) | Difference (%) |
|---|---|---|---|
| 100 | 12.18 | 12.13 | 0.41 |

Mwanga et al. [20] developed a Geometallurgical Comminution Test (GCT) that requires a small amount of initial sample and a jar mill (Capco, Ipswich, UK). The grinding test is performed on a sample under 3.35 mm with a starting weight of 220 g and can be performed within 2–3 h. The sample is ground while dry for 2, 5, 10, 17, and 25 min. After each grinding time, the PSD is determined by sieving, and the sample is returned to the mill for further grinding. $P_{80}$ is obtained from the PSD, and the power consumption is measured during the test. When the test is performed at a constant sample mass and mill

parameters (number of revolutions, grinding batch), it can be assumed that the energy supplied to the mill per unit time is constant. From Equation (1), for the given feed size, the change in specific grinding energy is proportional to the reciprocal of the square root of $P_{80}$:

$$W \cdot \frac{\sqrt{P_{80}}}{10} = \text{constant} \Rightarrow w_{i,GCT} = W \cdot \frac{\sqrt{P_{80}}}{100} \tag{28}$$

Comparing the results from the two grindability tests revealed that there is a linear relationship between the work indices. The model for estimating the Bond work index from the GCT test data is then given by Equation (29):

$$w_i = w_{i,GTC} \cdot \left( \frac{1}{\sqrt{\lambda}} \cdot \eta \cdot 1/4 \right) \tag{29}$$

wherein:

$\lambda$—geometric division factor and is $\lambda = 2.65$;

$\eta$—mill drive and engine efficiency and amounts to $\eta = 0.64$;

$W_{i,GTC}$—operating index of the GCT, calculated using Equation (28).

The authors stated that the test and performance of the presented method were confirmed on several ores, with the relative error ranging from 0.70% to f 8.8%. The advantages of this method are the small sample quantity that is needed (220 g of the initial sample) and the short testing time (results can be obtained in 25 min, taking the entire test no more than three hours). The disadvantage is the availability of the mill itself; the authors recognised that the proposed method does not aim to substitute the standard Bond test.

Lewis et al. [21] developed a new method of grinding testing based on computer simulation, closely related to the standard Bond method. The simulation is based on a mathematical algorithm that simulates a standard Bond test and is divided into two parts. The first part uses experimental data from the first grinding cycle to obtain the initial parameters of the model. The calculated parameters and raw material characteristics are stored in a database to be used in the second part of the simulation for prediction purposes. The prediction method simulates a standard test. For each grinding cycle, all raw material that is smaller than the opening of the comparative sieve is replaced by a representative mass of the starting sample. The calculation continues using the parameter values set for a given grinding cycle. Four grinding cycles are calculated automatically. A check is performed during the fourth and any subsequent grinding cycles to assess whether the newly formed undersize mass per mill revolution $G$ (g/rev) is constant (within 3%) for the last three grinding cycles. If $G$ (g/rev) is constant, a steady state is reached; otherwise, the computer procedure continues with the next grinding cycle. When a steady state is reached, the Bond work index is calculated using Equation (32). The mean square relative difference between the values of the Bond work index obtained by the standard method and the values obtained by computer simulation is 2.81%.

Aksani and Sönmez [22] proposed a computer simulation of the Bond grind test using a cumulative kinetic model [23,24]. The model contains only two parameters, which simplifies the interpretation of the results. Equation (30) gives the relationship between the comminution speed and the particle size:

$$k = C \cdot x^n \tag{30}$$

wherein:

$k$—breakage rate constant ($\text{min}^{-1}$);

$C$ and $n$—constants that are dependent on the mill and material characteristics;

$x$—sieve size ($\mu$m).

A standard Bond mill and a standard Bond grind test were used to determine the model parameters. The test is performed on a sample of mass $M$ (g) that is 700 cm$^3$ of size class $-3.35$ mm. The sample is ground at times of 0.5 min, 1 min, 2 min, and 4 min. After each grinding cycle, the analysis of the PSD is determined for the sample.

The PSD products are combined and returned to the mill for the subsequent grinding cycle. The grinding rate constant $k$ is calculated by nonlinear regression using the obtained cumulative reflection data in relation to the grinding time. To calculate the parameters $C$ and $n$, Equation (30) should be logarithmic, and then linear regression should be applied. The computer simulation uses PSD data, initial input mass, kinetic model parameters, and mill speed for the first grinding cycle. The prediction test simulates the standard Bond procedure. After each grinding cycle, the newly formed undersize mass per mill revolution $G$ (g/rev) is calculated, and the material under $P_{100}$ is replaced with the same mass of feed sample. The calculation continues until $G$ (g/rev) becomes constant for the last three grinding cycles. When a steady state is reached, the parameters obtained in the last grinding cycle and Equation (32) are used to calculate the Bond work index. The mean square relative error between the values of the Bond work index obtained by the standard method and the values obtained by computer simulation was 2.54%.

Ford and Sithole [25] provided an abbreviated method for $w_i$ estimation consisting of two tests. The first test was performed with only one grinding cycle, and the second test was performed with three grinding cycles.

In the first test, a sample of 700 cm$^3$ with a size of 100% under 3.35 mm is ground in a standard Bond ball mill in time intervals of 0.5 min, 1 min, 2 min, and 4 min. After each grinding run, the mass of the sample is measured, and the PSD is determined. These data are then used to calculate the parameter $k$ for each size x (see Equation (31)).

$$W_{(x,t)} = W_{(x,0)} \cdot \exp(-k \cdot t) \tag{31}$$

wherein:

$t$—grinding time (min);
$W_{(x,t)}$—cumulative content of screening aperture $x$ during grinding $t$;
$W_{(x,0)}$—cumulative reflection content of the initial sample for the sieve opening $x$;
$k$—breakage rate constant (min$^{-1}$).

The model describes a mathematical simulation in a closed grinding cycle. In the simulation, the number of revolutions varies until a circulating load of 250% is reached. The parameters $G$ (g/rev), $P_{80}$ (μm), and $F_{80}$ (μm) are estimated using simulation, and the Bond work index is estimated using the standard method Bond equation (Equation (32)) using the simulated parameters (Equation (33)).

$$w_i = \frac{44.5}{p_{100}{}^{0.23} \cdot (G)^{0.82} \cdot \left( \frac{10}{\sqrt{P_{80}}} - \frac{10}{\sqrt{F_{80}}} \right)} \tag{32}$$

$$w_{i,FS1} = \frac{44.5}{p_{100}{}^{0.23} \cdot (G_s)^{0.82} \cdot \left( \frac{10}{\sqrt{P_{80,s}}} - \frac{10}{\sqrt{F_{80,s}}} \right)} \tag{33}$$

wherein $G_s$, $F_{80,s}$, and $P_{80,s}$ are $G$ (g/rev), $P_{80}$ (μm), and $F_{80}$ (μm) obtained by simulation, respectively.

The feature of this method is that the work indices can be simulated for different PSD based on the results of only one grinding cycle.

The second proposed test is based on the standard Bond test, considering only the first three grinding cycles. After the third cycle, $G$ (g/rev) and $P_{80}$ are determined and used to calculate the Bond work index via Equation (34):

$$w_{i,FS2} = \frac{44.5}{p_{100}{}^{0.23} \cdot (G_3)^{0.82} \cdot \left( \frac{10}{\sqrt{P_{80,3}}} - \frac{10}{\sqrt{F_{80,3}}} \right)} \tag{34}$$

wherein $G_3$, $F_{80,3}$, and $P_{80,3}$ are $G$ (g/rev), $P_{80}$ (μm), and $F_{80}$ (μm) obtained experimentally after only three cycles of the Bond standard test.

The mean square relative error between $w_i$ and $w_{i,FS1}$, and $w_i$ and $w_{i,FS2}$ resulted in 11.71%, and 2.20%, respectively. The second procedure takes longer but leads to better results than the first one.

Gharehgheshlagh [26] presented a method for calculating the Bond work index that tracks the grinding kinetics in a Bond ball mill. The method is fast and practical because it establishes a relationship between the grinding parameters and the parameters of the Bond equation and eliminates specific steps of the laboratory test due to the reduction of the grinding cycle. The test is performed by grinding 700 cm$^3$ of a sample 100% under 3.35 mm in a Bond ball mill for 0.33, 1, 2, 4, and 8 min. After each grinding cycle, the grinding product PSD is determined and returned to the mill for the subsequent grinding cycle. This grinding kinetics analysis is used to determine the functional dependence between the number of mill revolutions and undersize mass passing $P_{100}$ (m$_{us}$) as well as the relationship between the number of mill revolutions and $P_{80}$ (μm) using the least-squares numerical method. The first function determines the number of mill revolutions $N_{250\%}$ (revolutions) required to obtain the under-size mass, such that the circulating load is 250%. Based on the values of $N_{250\%}$ (revolutions) and the determined functional dependencies, the parameters $G$ (g/rev) and $P_{80}$ (μm) are estimated, and Equation (32) can be used to estimate the work index. The mean square relative error between the real and estimated $w_i$ was 1.23%.

Ciribeni et al. [27] introduced a Bond test simulation based on the cumulative kinetic model [23,24]. The simplified procedure consists of calculating the kinetic parameters after only one grinding run, instead of a series of runs. Finally, the estimation of $w_i$ is performed through mathematical simulation. The test is performed by grinding a 700 cm$^3$ sample, 100% under 3.35 mm, in a Bond ball mill for 5 min. The kinetic parameters are determined: $k$ by Equation (30) and $C$ and $n$ by Equation (35).

$$\ln(k) = \ln(C) + n \cdot \ln(x) \tag{35}$$

Once estimated by the simulation of $G$ and $P_{80}$, the Bond work index is estimated using Equation (32). Several ores were used for validation, and the mean square relative error between standard and calculated work index was reported as 6.31%.

Magdalinović [28] presented an abbreviated test for determining the work index based on performing two grinding cycles and relying on the law of first-order kinetics. The test is performed on a sample prepared 100% under 3.35 mm, in a standard Bond mill. Feed sample PSD is obtained, the initial sample mass of 700 cm$^3$, $M$ (g), is determined, and the grinding product mass at steady-state, $IGP$, is calculated with Equation (36):

$$IGP = M/3.5 \; [\text{g}] \tag{36}$$

The first grinding cycle feed is prepared with the $IGP$ weight of the initial sample and made equal to $M$ weight with the initial sample after removing by sieving the undersize of $P_{100}$. This composite sample is ground for an arbitrary number of mill revolutions ($N_1$, usually 50, 100, or 150), and the oversize mass and the undersize mass are weighed. The oversize grinding rate constant ($k$) can be calculated using Equation (37);

$$k = n \cdot \frac{\ln R_0 - \ln R_1}{N_1} \tag{37}$$

wherein:

$R_0$—oversize in the initial sample (%);
$R_1$—oversize in the product of cycle 1 (%);
$n$—number of revolutions per min, (min$^{-1}$);
$N_1$—total number of mill revolutions in cycle 1.

Once the grinding rate constant $k$ is determined, it is used to obtain the necessary mill revolutions $N_2$ required to obtain a circulating load of 250%. The second cycle feed is obtained as it was in the first cycle. The second cycle involves a grinding operation for

$N_2$ revolutions. Once the cycle is finished, the product is sieved at $P_{100}$. The undersize mass should be approximately *IGP*. The G value (g/rev) is calculated, and $P_{80}$ is obtained from the PSD. Equation (32) can be used to estimate the Bond work index. The mean square relative error between the actual and the calculated values of the work index by the Magdalinović test with two grinding cycles was 4.9%. In 2003, Magdalinović [29] proposed the abbreviated test with three grinding cycles by adding one cycle to the procedure proposed in 1989. In this case, the mean square relative error diminished to 2.75%. As expected, a lower minor error is obtained with the abbreviated three cycle test.

Todorovic et al. [30] proposed an abbreviated method that could be done with two, three, or four grinding cycles. Each grinding cycle is done in the same way as in the standard Bond procedure. In the shortened procedure with two grinding cycles, the PSD of the initial sample, $F_{80}$ (µm), and X (the oversize mass at $P_{100}$) are determined. From the feed, which must be prepared 100% under 3.327 mm, a sample of 700 cm$^3$ is taken, and its weight M (g) is determined. This sample is ground in a Bond ball mill for an arbitrary number of revolutions ($N_1$ = 50, 100, or 150 revolutions). Afterwards, the grinding product is sieved at $P_{100}$ and R (retained weight, g) and D (undersize weight, g) are determined. The undersize weight D is the sum of the undersize in the fresh feed $D_u$ (g) and the newly formed undersize $D_n$. The newly formed undersize mass $D_n = D - D_u$ is calculated. In the first cycle, $D_u = M \cdot (1 - X)$ (g), while in the subsequent cycle, $D_u = D_{n-1} \cdot (1 - X)$ (g), wherein $D_{n-1}$ is the undersize mass of the sieves from the previous cycle, (g). The newly formed undersize mass per mill revolution G (g/rev) is then calculated, and the number of mill revolutions for the subsequent grinding cycle $N_n$ is according to Equation (38).

$$N_n = \frac{\frac{M}{3.5} - D_{(n-1)} \cdot (1 - X)}{G} \ [\text{rev}] \tag{38}$$

A fresh feed sample equaling $D_{n-1}$ is blended with the retained material from the previous cycle, fed into the mill, and ground for $N_n$ revolutions. The grinding product is again sieved, and the retained product is weighed to obtain the R of this cycle. The constant k is then calculated using Equation (39):

$$k = \frac{n \cdot (\ln R_0 - \ln R)}{N} = \frac{n \cdot \left[ \ln \left( \frac{R_{(n-1)}}{M} + \frac{D_{(n-1)}}{M} \cdot X \right) - \ln \frac{R}{M} \right]}{N} \tag{39}$$

The required number of revolutions N is calculated to produce the steady-state weight undersize at 250% circulating load (Equation (40)):

$$N = \frac{n}{k} \left[ \ln \left( \frac{2.5}{3.5} \cdot 100 + \frac{X}{3.5} \cdot 100 \right) - \ln \left( \frac{2.5}{3.5} \cdot 100 \right) \right] \tag{40}$$

The parameter G (g/rev) is calculated using Equation (41), and the last cycle $G_e$ value is estimated using Equation (42). The $P_{80}$ value in this cycle is estimated using Equation (43).

$$G = \frac{\frac{M}{3.5} \cdot X}{N} \tag{41}$$

$$G_e = 1.158 \cdot G \ (\text{g/rev}) \tag{42}$$

$$P_{80} = 1.035 \cdot P_{80,n-1} \ (\text{µm}) \tag{43}$$

Using the values of $G_e$ (g/rev) and $P_{80}$ (µm) in Equation (32), the estimation of the work index $w_{i,T}$ (kWh/t) is obtained. Table 8 shows the results reported by Todorovic et al. [30] on mixtures of limestone and andesite, comparing $w_i$ with $w_{i,T}$ obtained by the abbreviated test with two, three, and four grinding cycles. The mean square relative error ranged from 3.0% to 5.2%.

**Table 8.** Comparative values of the Bond work index according to the standard Bond test and the abbreviated test by Todorovic et al.

| Sample | $P_{100}$ (μm) | $w_i$ (kWh/t) | $w_{i,T}$ (kWh/t) | | | Difference (%) | | |
|---|---|---|---|---|---|---|---|---|
| | | | II Cycle | III Cycle | IV Cycle | II Cycle | III Cycle | IV Cycle |
| Limestone:Andesite 0:100 | 74 | 18.09 | 17.53 | 17.37 | 17.90 | −3.09 | −3.98 | −1.05 |
| Limestone:Andesite 25:75 | 74 | 17.03 | 17.69 | 16.73 | 16.80 | 3.87 | −1.75 | −1.33 |
| Limestone:Andesite 50:50 | 74 | 15.15 | 15.58 | 14.89 | 15.03 | 0.51 | −3.93 | −3.02 |
| Limestone: Andesite 75: 25 | 74 | 14.51 | 14.39 | 13.86 | 14.03 | −0.82 | −4.48 | −3.34 |
| Limestone:Andesite 100:0 | 74 | 13.90 | 14.50 | 15.14 | 14.53 | 4.32 | 8.93 | 4.52 |
| Mean relative square error | | | | | | 2.97 | 5.18 | 2.92 |

A summary of the mean relative square error reported by the authors in each proposal is depicted in Figure 2. Considering the relative error values and the simplification of the laboratory procedure, the method proposed by Horst and Bassarear and the one by Gharehgheshlagh are advantageous.

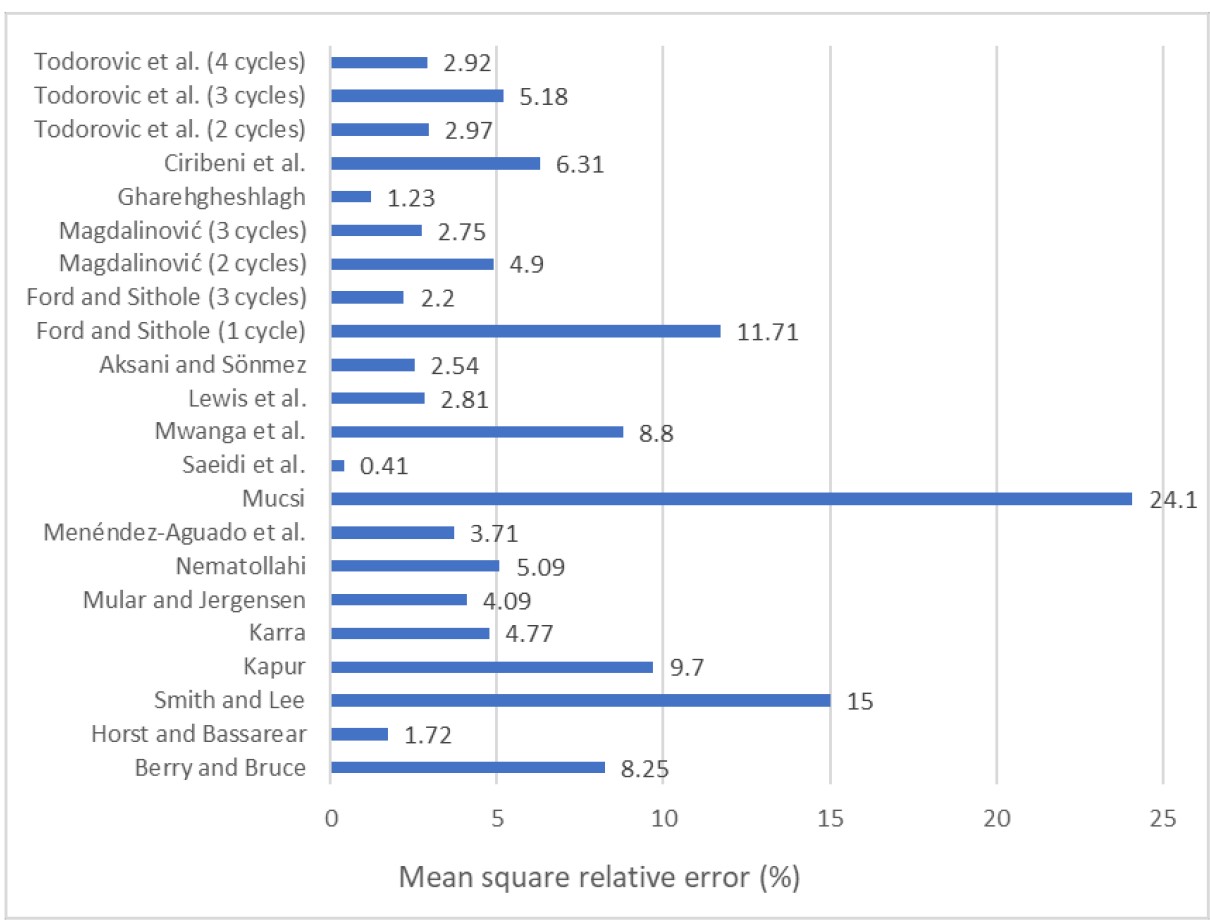

**Figure 2.** Summary of relative errors of alternative procedures.

## 3. Conclusions

Alternative abbreviated and simplified procedures for determining the work index have been proposed through the years. This review presented alternative shorter, simplified, and faster procedures that can be classified into two groups:

1. Alternative tests that simulate the standard Bond test with an abbreviated procedure;
2. Alternative tests based on determining problem sample grindability using a reference sample with $w_i$ known.

Alternative tests from the first group are based on the use of a Bond standard ball mill for the reach the steady-state more quickly [28–30] or for performing the mathematical simulation of the standard test [13,14,21,22,27].

Alternative tests from the second group can be performed in a different mill, usually needing less sample than the standard procedure. All of the methods aim to give a close estimation of the Bond work index when the standard Bond ball mill is not available and are faster procedures with a reduced number of grinding steps. The longest alternative test requires 3–4 grinding cycles, while the shortest one can be performed with one grinding cycle. It must be considered that the standard procedure compels a minimum of 5 grinding cycles, with 7–10 grinding cycles usually being necessary.

In general, the mean square error data presented cannot be understood as a validity indicator, for in some cases, the reported value was based on just a few tests or with few ores. However, these data indicate that shorter procedures (i.e., with just one grinding cycle) are usually less reliable, yielding a higher mean square error. Nevertheless, due to the advantage in laboratory time, they could be recommended if ore feed is the same, which could be the case of the periodic grindability control in a specific mine.

Finally, after an adequate grinding kinetic behaviour characterisation of the ore, alternative tests based on the simulation of the standard Bond test could be recommended when considering the process digitalisation as part of the global digitalisation strategy in the mining industry.

**Author Contributions:** Conceptualisation, V.N. and J.M.M.-A.; methodology, G.G.G. and J.M.M.-A.; investigation, V.N., M.S.T., G.G.G. and A.L.C.-V.; resources, V.N., M.T., M.S.T. and J.M.M.-A.; writing—original draft preparation, V.N., M.T. and M.S.T.; writing—review and editing, A.L.C.-V., G.G.G. and J.M.M.-A.; visualisation, V.N., M.S.T., G.G.G. and J.M.M.-A.; supervision, M.T. and A.L.C.-V. All authors have read and agreed to the published version of the manuscript.

**Funding:** This research did not receive external funding.

**Institutional Review Board Statement:** Not applicable.

**Informed Consent Statement:** Not applicable.

**Data Availability Statement:** Not applicable.

**Conflicts of Interest:** The authors declare no conflict of interest.

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
