# Peer review of "A Review of Alternative Procedures to the Bond Ball Mill Standard Grindability Test"

_metals, doi:10.3390/met11071114_

Round 1

Reviewer 1 Report

A very focused and highly relevenat contribution to the area of ball milling and grandibility.

The review suggests that a spell/grammar check be conducted prior to final submission.

Author Response

R: A very focused and highly relevenat contribution to the area of ball milling and grandibility. The review suggests that a spell/grammar check be conducted prior to final submission.

A: Authors appreciate the reviewer comment. The revised version of the manuscript has an additional English revision.

Reviewer 2 Report

I can provide the following comments to improve both the presentation and content of this paper:

1) Include in the abstract the main conclusion of this survey.

2) Expand the introduction referencing other alternative methods for estimating the grindability of the material. Add also comments concerning the advantages of using the Bond Index. It will nice to include some historical material about this test.

3)Summarize the finding in a table with all the surveyed methods indicating their main operational characteristics, weak and strong points. Similar to the ones found in [1] and [2].

4) A thorough comparison and a critical analysis of the different tests should be provided; as done for instance in [3].

5) In the conclusions the authors recommend the use of tests based on the simulation of the standard Bond test, but no information is given to support this recommendation. 

Additional references

[1] https://www.sgs.com/-/media/global/documents/technical-documents/sgs-min-tech-pub-2005-06-small-scale-tests-lr-en-11-09-v1.pdf

[2] Mwanga A, Rosenkranz J, Lamberg P. Testing of Ore Comminution Behavior in the Geometallurgical Context—A Review. Minerals. 2015; 5(2):276-297. https://doi.org/10.3390/min5020276

[3] Valerevich Lvov, Vladislav and Sergeevich Chitalov, Leonid, Comparison of the Different Ways of the Ball Bond Work Index Determining (September 12, 2019). International Journal of Mechanical Engineering and Technology, 10(3), 2019, pp. 1180-1194, Available at SSRN: https://ssrn.com/abstract=3452642

Author Response

R: I can provide the following comments to improve both the presentation and content of this paper:

1) Include in the abstract the main conclusion of this survey.

A: Thank you for the suggestion, it has been included in the revised manuscript.

R: 2) Expand the introduction referencing other alternative methods for estimating the grindability of the material. Add also comments concerning the advantages of using the Bond Index. It will nice to include some historical material about this test.

A: Authors appreciate reviewer suggestion. Some reference to impact breakage tests were included, specially drop weight test.

R: 3)Summarize the finding in a table with all the surveyed methods indicating their main operational characteristics, weak and strong points. Similar to the ones found in [1] and [2].

A: Thank you for the suggestion, this Table has been elaborated and included as supplementary material.

R: 4) A thorough comparison and a critical analysis of the different tests should be provided; as done for instance in [3].

A: Thank you for the suggestion. The reference 3 was included in the introduction section.

R: 5) In the conclusions the authors recommend the use of tests based on the simulation of the standard Bond test, but no information is given to support this recommendation.

A: Thank you for the comment. The recommendation was rewritten as follows: “alternative tests based on the simulation of the standard Bond test, after an adequate grinding kinetic behaviour characterisation of the ore, could be recommended when considering the process digitalisation as part of the global digitalisation strategy in the mining industry”

Additional references

[1] https://www.sgs.com/-/media/global/documents/technical-documents/sgs-min-tech-pub-2005-06-small-scale-tests-lr-en-11-09-v1.pdf

[2] Mwanga A, Rosenkranz J, Lamberg P. Testing of Ore Comminution Behavior in the Geometallurgical Context—A Review. Minerals. 2015; 5(2):276-297. https://doi.org/10.3390/min5020276

[3] Valerevich Lvov, Vladislav and Sergeevich Chitalov, Leonid, Comparison of the Different Ways of the Ball Bond Work Index Determining (September 12, 2019). International Journal of Mechanical Engineering and Technology, 10(3), 2019, pp. 1180-1194, Available at SSRN: https://ssrn.com/abstract=3452642

Reviewer 3 Report

I have read the manuscript with a lot of interest. However, I believe major revision is required before the submission is considered for publication.

Below are some of the comments for the benefits of the authors:

The abstract does not capture the highlights of the review paper. Authors should make an effort to summarily present the objective of the paper, the major schools of thought around the Bond grindability tests, and the gaps or potential avenues for further research around the Bond grindability test.

Morrell, Austin et al. (1984) [[10] L.G. Austin, R.R. Klimpel, P.T. Luckie, Process engineering of size reduction: Ball milling, Society of Mining Engineers of the AIME, New York, 1984] and other researchers have discussed at length the shortcomings of Bond’s procedure. Stating that the procedure is reliable is misleading when we know from available literature that this is not true. Yes, the Bond index is useful and widely accepted in the mining industry but it has serious limitations. This needs to be captured accurately in the Introduction section.

It also needs to be clear whether the review is around shortening the duration of the current testing procedure and doing away with the standard mill or the review looks at other aspects. It is not clear to me what research methodologies were used to select the most relevant papers in line with the objective of the submitted review paper.

The authors need to first review the original procedure proposed by Bond and discuss its merits and shortcomings. They can then present the procedure by Berry and Bruce. I find it difficult to comprehend the procedure by Berry and Bruce. Very little description is made by the authors to enable readers a fuller appreciation of the procedure. I recommend the authors to explain in detail the procedure in such a way that anyone can follow the description and perform the procedure without a problem.

I also see that a mill of diameter 305 mm is required, yet the authors argued in the introduction to be looking for alternatives that may do away with the standard mill by Bond. I therefore recommend that Sections 1 and 2 be aligned properly.

In Equations (1) and (2), it is not clear how long one needs to mill the sample before they can measure the 80% passing size and the specific power consumption. I also need clarity on where the Bond index of the reference ore is taken from. This is because the reference ore needed brings in another layer of complexity to the alternative method that does not exist with Bond’s procedure.

The starting feed is less than 1.651 mm in the procedure by Berry and Bruce. But what should the particle size distribution be? Are they any requirements here or anything is good for as long as the feed is less than 1651 mm? What are the implications of testing different size distributions of the same ore on the final “Bond” index?

The authors argue that “The validity of this procedure depends on the accuracy of stopping the unknown sample grinding test after a specific power consumption and on the similarity of the particle size distribution (PSD) of the feed samples.” How do you ensure this takes place? Is this not far more challenging to do than Bond’s procedure? Again, because the introduction is not clear in terms of the aim of this review, it becomes difficult to get a fair appreciation of the paper submitted. I am finally not clear how the procedure proposed by Berry and Bruce is fast with no little information on the technicalities of the procedure.

The procedure by Horst and Bassarear seems to be interesting but there are also serious problems.

First of all, the authors explain that “A reference ore sample weighing 1 kg is ground until the desired PSD of the mill is obtained”. How do you ensure that you are able to get to the desired product size distribution and what is that product size distribution? Secondly, the description is also difficult to comprehend and should be improved both in terms of the grammar and of the content which is quite shallow. Thirdly, a worked example of how the procedure by Horst and Bassarear would really enhance the text. Lastly, several references are needed in support of the respective errors associated with the above two techniques (i.e. 8.25 % and 1.72 %) should be provided as well as corresponding experimental conditions and ore tested.

Smith and Lee propose a viable alternative; however, the authors should explain how open-grinding tests are done in great detail. Examples and supporting references where the procedure by Smith and Lee was compared to Bond’s should be covered. In terms of duration of the tests, can we say that this technique is shorter than Bond’s? Do we use any mill, any feed size distribution? What are the testing conditions that one needs to adhere to in order to get the best estimates of the Bond index? If this procedure compares to Bond’s to within 15 % discrepancy, why is it not finding why use? Do you have references encouraging the use of this technique besides Smith and Lee (1968)?

In the method by Kapur (1970), what does Pc stand for? Here also, please unpack the description for the benefit of readers and perhaps use a worked example supported with appropriate references to strengthen the review.

The procedures proposed by Karra (1980) and by Mular and Jergensen (1982) are quite well explained but there is still room for improvement. Most of the comments made above also apply here. I recommend that each alternative procedure to be covered in a separate dedicated section so that you may have so many sections there are procedures besides the Introduction and the Conclusion sections. An additional section before the conclusion dealing with the strengths and shortcomings of each technique is needed. This particular section should identify gaps with existing techniques that will constitute avenues for future research. A comparison of the accuracy and precision of all the reviewed techniques should also be covered in that section.

Procedures by Nematollahi (1994) and by Menéndez-Aguado et al. (2005) are well covered. But major overhaul is needed for the remaining procedures following the comments made earlier. The biggest issue is the lack of detailed description, illustrations, and supporting references. In reviewing the manuscript, the authors should make sure the work is aligned with the objective of this review. The challenge is that the objectives are not yet clearly articulated and it is not clear to me for the most part whether I can reproduce the procedures presented in this manuscript. And for me to do the latter, I may probably need to look for the original papers by the various researchers which diminishes the value of the review. This serious shortcoming needs to be addressed if the review paper is to be published in this journal.

The key question in my view that was not addressed is to recommend a technique from those reviewed in substitution for Bond’s and the reason for such a choice.

My last comment is that very few references were used in compiling this manuscript. Because it is a review, one would expect anything above at least 60 references. I recommend the authors to build a strong review by scanning quite widely through the existing body of knowledge. It also needs to be state that the scope of work covers ball mills only amongst others.

I believe that once the authors work through the comments made for major revision, the paper can be considered for publication.

Author Response

R: I have read the manuscript with a lot of interest. However, I believe major revision is required before the submission is considered for publication.

Below are some of the comments for the benefits of the authors:

The abstract does not capture the highlights of the review paper. Authors should make an effort to summarily present the objective of the paper, the major schools of thought around the Bond grindability tests, and the gaps or potential avenues for further research around the Bond grindability test.

A: Thank you for the comment, the abstract has been changed in the revised version; the introduction also includes now references to impact tests as a different approach to grindability characterisation

R: Morrell, Austin et al. (1984) [[10] L.G. Austin, R.R. Klimpel, P.T. Luckie, Process engineering of size reduction: Ball milling, Society of Mining Engineers of the AIME, New York, 1984] and other researchers have discussed at length the shortcomings of Bond’s procedure. Stating that the procedure is reliable is misleading when we know from available literature that this is not true. Yes, the Bond index is useful and widely accepted in the mining industry but it has serious limitations. This needs to be captured accurately in the Introduction section.

A: Thank you for the comment. The paper is not aimed to be a blind defence of Bond’s methodology, the authors have more than 25 years of experience working on Bond’s methodology and are fully aware of the procedure shortcomings. The affirmations have been carefully reviewed, and the objective is clearly defined.

R: It also needs to be clear whether the review is around shortening the duration of the current testing procedure and doing away with the standard mill or the review looks at other aspects. It is not clear to me what research methodologies were used to select the most relevant papers in line with the objective of the submitted review paper.

A: The selection of alternative procedures was based on the practical experience of the laboratories involved in this research, as stated in the revised version of the manuscript.

R: The authors need to first review the original procedure proposed by Bond and discuss its merits and shortcomings. They can then present the procedure by Berry and Bruce. I find it difficult to comprehend the procedure by Berry and Bruce. Very little description is made by the authors to enable readers a fuller appreciation of the procedure. I recommend the authors to explain in detail the procedure in such a way that anyone can follow the description and perform the procedure without a problem.

A: Thanks for the deep revision of the paragraphs about the Berry and Bruce methodology. This is probably the best known and most widely spread alternative procedure to Bond test, with more references availability to the readers, so it was described

R: I also see that a mill of diameter 305 mm is required, yet the authors argued in the introduction to be looking for alternatives that may do away with the standard mill by Bond. I therefore recommend that Sections 1 and 2 be aligned properly.

A: Thank you for the comment. The reviewer is right. It was adequately corrected in the revised version.

R: In Equations (1) and (2), it is not clear how long one needs to mill the sample before they can measure the 80% passing size and the specific power consumption. I also need clarity on where the Bond index of the reference ore is taken from. This is because the reference ore needed brings in another layer of complexity to the alternative method that does not exist with Bond’s procedure.

A: Authors totally agree with the reviewer comments; these are perhaps the main difficulties of this method. The grinding time depends on the reading on a power-meter stop the grinding test with the ore of unknown grindability at a specific value; this introduces a source of variability. Regarding the need of a reference ore, it is the sign that Berry and Bruce accepted the Bond grindability approach, but the understanding was not deep enough, for a specific ore always shows variations in Bond work index at different P100 sieve sizes.

R: The starting feed is less than 1.651 mm in the procedure by Berry and Bruce. But what should the particle size distribution be? Are they any requirements here or anything is good for as long as the feed is less than 1651 mm? What are the implications of testing different size distributions of the same ore on the final “Bond” index?

The authors argue that “The validity of this procedure depends on the accuracy of stopping the unknown sample grinding test after a specific power consumption and on the similarity of the particle size distribution (PSD) of the feed samples.” How do you ensure this takes place? Is this not far more challenging to do than Bond’s procedure? Again, because the introduction is not clear in terms of the aim of this review, it becomes difficult to get a fair appreciation of the paper submitted. I am finally not clear how the procedure proposed by Berry and Bruce is fast with no little information on the technicalities of the procedure.

A: Thanks for the thorough revision of the paragraphs about the Berry and Bruce methodology.

R: The procedure by Horst and Bassarear seems to be interesting but there are also serious problems. First of all, the authors explain that “A reference ore sample weighing 1 kg is ground until the desired PSD of the mill is obtained”. How do you ensure that you are able to get to the desired product size distribution and what is that product size distribution?

A: Thank you for the comment, it is explained in the first sentence that this test needs some grinding kinetics knowledge, which is acquired with the test results. Once characterised the kinetic model, a particular R value at a specific x size can be obtained with a t grinding time.

R: Secondly, the description is also difficult to comprehend and should be improved both in terms of the grammar and of the content which is quite shallow.

A: Additional details were included in the test description.

R: Thirdly, a worked example of how the procedure by Horst and Bassarear would really enhance the text.

A: Thank you for the suggestion. Authors consider that the explanation has been improved so it can be replied easily. However, the original reference can be easily found.

R: Lastly, several references are needed in support of the respective errors associated with the above two techniques (i.e. 8.25 % and 1.72 %) should be provided as well as corresponding experimental conditions and ore tested.

A: Authors fully agree with the reviewer comment, and it was included in the paragraph that these values were reported by the authors (Horst and Bassarear)

R: Smith and Lee propose a viable alternative; however, the authors should explain how open-grinding tests are done in great detail. Examples and supporting references where the procedure by Smith and Lee was compared to Bond’s should be covered.

A: This methodology is based in the estimation of final gbp from the gbp obtained from cycle 1. It is pointed clearly that the number of mill revolutions should be less than 300, which applies usually with ores with expected work index below 15-17 kWh/t

R: In terms of duration of the tests, can we say that this technique is shorter than Bond’s? Do we use any mill, any feed size distribution? What are the testing conditions that one needs to adhere to in order to get the best estimates of the Bond index? If this procedure compares to Bond’s to within 15 % discrepancy, why is it not finding why use? Do you have references encouraging the use of this technique besides Smith and Lee (1968)?

A: Thank you for the points; it was included that this test is performed in the standard mill, under the standard test conditions, as already mentioned. It is of course quite shorter than the standard test, because only is performed the first cycle, as explained. Under the authors experience, the influence of feed particle size, which had only the restriction of being under 3.35 mm conditions greatly the results, especially in the initial cycles. A comment following this argument has been included in the revised version. Finally, most of the references use this article as a source of Bond work index values, not as a recommended methodology. However, the first published discussion of this paper was signed by Fred Bond, and he valuated greatly the time saving procedure, recommending its use in quick daily control of the grinding feed properties.

R: In the method by Kapur (1970), what does Pc stand for? Here also, please unpack the description for the benefit of readers and perhaps use a worked example supported with appropriate references to strengthen the review.

A: Sorry for the mistake, it was corrected to Pk. The suggestion of an example would be really good for any of the methods, but authors prefer to keep a short but precise description of the selected methodologies. When the critical analysis can help the reader to choose one methodology or another, it is highlithed.

R: The procedures proposed by Karra (1980) and by Mular and Jergensen (1982) are quite well explained but there is still room for improvement. Most of the comments made above also apply here. I recommend that each alternative procedure to be covered in a separate dedicated section so that you may have so many sections there are procedures besides the Introduction and the Conclusion sections. An additional section before the conclusion dealing with the strengths and shortcomings of each technique is needed. This particular section should identify gaps with existing techniques that will constitute avenues for future research. A comparison of the accuracy and precision of all the reviewed techniques should also be covered in that section.

A: Thank you for the comments. Again, this suggestion would be good if any of the methodologies would have the same potential of benefit against the standard method. Some of the methods are almost just described because they are not generally well known but they pose some interesting approach which can help the readers to even find new alternatives. This is the case of the referred methods in this comment.

R: Procedures by Nematollahi (1994) and by Menéndez-Aguado et al. (2005) are well covered. But major overhaul is needed for the remaining procedures following the comments made earlier. The biggest issue is the lack of detailed description, illustrations, and supporting references. In reviewing the manuscript, the authors should make sure the work is aligned with the objective of this review. The challenge is that the objectives are not yet clearly articulated and it is not clear to me for the most part whether I can reproduce the procedures presented in this manuscript. And for me to do the latter, I may probably need to look for the original papers by the various researchers which diminishes the value of the review. This serious shortcoming needs to be addressed if the review paper is to be published in this journal.

A: Dear author, the subject covered by this review is so wide that of course it is needed to look for the original papers; furthermore, authors intention is not to avoid that but to estimulate the revision of that literature, for many work was made in the past by researchers, most of it easily available today, and not so deeply understood.

R: The key question in my view that was not addressed is to recommend a technique from those reviewed in substitution for Bond’s and the reason for such a choice.

A: Thank you for the comment, in the revised version some recommendations have been made.

R: My last comment is that very few references were used in compiling this manuscript. Because it is a review, one would expect anything above at least 60 references. I recommend the authors to build a strong review by scanning quite widely through the existing body of knowledge. It also needs to be state that the scope of work covers ball mills only amongst others.

I believe that once the authors work through the comments made for major revision, the paper can be considered for publication.

A: Dear reviewer, thank you very much for your comments, which helps to improve our paper. We respect your point of view regarding the number of references; really it would be easy for us to increase it, even with our own production related to this subject; however, it was our intention to include only a selected number of reference, being the selection made truly as a relevant part of this work. If readers take the time to look up those references they for sure would enrich their knowledge about the Bond work index alternatives.

Reviewer 4 Report

Dear Authors,

The manuscript is interesting as it reviews alternative procedures to the Bond grindability test.

Please find below some recommendations.

Introduction section

The paper will benefit if authors provide in the end of introduction section a paragraph or two underlining an importance of the topic and reasons for this review.

Please provide a description for ISO and ASTM in line 26 and other abbreviations throughout the paper.

The paper might benefit and attract more readers if authors summarize main findings of their review in a table comparing pros and cons of different grindability tests.

Line 483:

Was something missed after comma?

Lines 487-489:

The paper will benefit if authors exclude citations from conclusion section and add more explonation instead.

Line 492: ...quicker procedures...

Lines 493, 494: The longest alternative test requires four grinding cycles, while the  shortest one can be performed with one grinding cycle.

Maybe to add numbers or range of values?

Line 497: …for instance…

Readers will appreciate if authors provide reasons for choosing of this example or paraphrasing a sentence.

Regards,

Reviewer

Author Response

Dear Authors,

The manuscript is interesting as it reviews alternative procedures to the Bond grindability test.

Please find below some recommendations.

R: Introduction section

The paper will benefit if authors provide in the end of introduction section a paragraph or two underlining an importance of the topic and reasons for this review.

A: Authors appreciate the reviewer suggestion, it was included in the revised version.

R: Please provide a description for ISO and ASTM in line 26 and other abbreviations throughout the paper.

A: Thank you, it was revised and all abbreviations were described.

R: The paper might benefit and attract more readers if authors summarize main findings of their review in a table comparing pros and cons of different grindability tests.

A: Thank you for the suggestion, it was included as a supplementary material.

R: Line 483: Was something missed after comma?

A: Thank you for the comment, the mistake was corrected.

R: Lines 487-489: The paper will benefit if authors exclude citations from conclusion section and add more explonation instead.

A: Thank you for the comment. Authors agree that new references should not be added in the conclusion section. However, in this case all the references have been previously mentioned and addressed in section 2, so we prefer to use the reference as the way of addressing them.

R: Line 492: ...quicker procedures...

A: Thank you, it was changed by “faster procedure”

R: Lines 493, 494: The longest alternative test requires four grinding cycles, while the  shortest one can be performed with one grinding cycle. Maybe to add numbers or range of values?

A: OK it was changed by “The longest alternative test requires 3-4 grinding cycles, while the shortest one can be performed with one grinding cycle. It must be considered that the standard procedure compels a minimum of 5 grinding cycles, usually being necessary 7-10 grinding cycles”

R: Line 497: …for instance…

A: Thank you, changed by “i.e.”

R: Readers will appreciate if authors provide reasons for choosing of this example or paraphrasing a sentence.

A: Thank you for the suggestion. It was included: “Nevertheless, due to the advantage in laboratory time, they could be recommended if ore feed is the same, as could be the case of the periodic grindability control in a specific mine”.

Regards,

Reviewer

Round 2

Reviewer 2 Report

I thank the authors for taking into account all my suggestions and submitting  this improved version of their manuscript.

Reviewer 3 Report

After reading through the revised manuscript and the response by the authors, I believe now that the work can be accepted for publication.

I look forward to follow-up papers from the current. I hope these will now validate the best procedures identified for the measurement of the Bond index.